# Effects of High Intensity Laser Therapy in the Treatment of Tendon and Ligament Injuries in Performance Horses

**DOI:** 10.3390/ani10081327

**Published:** 2020-07-31

**Authors:** Paulina Zielińska, Jakub Nicpoń, Zdzisław Kiełbowicz, Maria Soroko, Krzysztof Dudek, Daniel Zaborski

**Affiliations:** 1Department of Surgery, Faculty of Veterinary Medicine, Wroclaw University of Environmental and Life Sciences, Plac Grunwaldzki 51, 50-366 Wrocław, Poland; jakub.nicpon@upwr.edu.pl (J.N.); zdzislaw.kielbowicz@upwr.edu.pl (Z.K.); 2Department of Horse Breeding and Equestrian Studies, The Faculty of Biology and Animal Science, Wroclaw University of Environmental and Life Sciences, Kożuchowska 5A, 51-161 Wrocław, Poland; maria.soroko@upwr.edu.pl; 3Faculty of Mechanical Engineering, Wroclaw University of Technology, Łukasiewicza 7/9, 50-231 Wrocław, Poland; krzysztof.dudek@pwr.edu.pl; 4Department of Ruminants Science, West Pomeranian University of Science and Technology, Doktora Judyma 10, 71-466 Szczecin, Poland; daniel.zaborski@zut.edu.pl

**Keywords:** horse, physiotherapy, high intensity laser therapy, ultrasound

## Abstract

**Simple Summary:**

High intensity laser therapy (HILT) has been introduced as a non-invasive therapy for musculoskeletal diseases in horses, but little is known about the impact of HILT in the treatment of tendon and ligament injuries. The aim of this study was to evaluate the effects of HILT on tendon and ligament injury treatment in horses. Twenty six horses with tendinopathies and desmopathies were randomly assigned to a HILT treated or to a non-treated group. Each horse from the treatment group underwent a series of fifteen HILT treatments. Clinical and ultrasound assessments were carried out. Clinical evaluation included: pain, swelling and lameness of the affected limb. The ultrasound examination evaluated lesion echogenicity and lesion percentage. In our study, HILT promoted analgesic and anti-oedema effects, with visual lameness reduction in horses with tendon and ligament injuries, and reduced lesion percentage but did not influence change in lesion echogenicity. HILT appears justifiable for its anti-inflammatory effects and can be used as a physiotherapeutic technique for supportive treatment of tendon and ligament injuries in horses. The introduction of laser therapy to veterinary medicine, particularly to analgesic therapy in horses, gives hope of improving the quality of life of patients with chronic musculoskeletal pain.

**Abstract:**

The aim of this study was to evaluate the effects of high intensity laser therapy (HILT) on tendon and ligament injury treatment in horses. Twenty six horses with tendinopathies were randomly assigned to a HILT treated or to a non-treated group. Each horse was subjected to the same rehabilitation programme. Horses from the treatment group underwent a series of fifteen HILT treatments with the same parameters. Clinical and ultrasound assessments were performed by the same veterinarian and were carried out before (day 0), during (day 13–15) and after treatment (day 38–40). Clinical evaluation included: pain, swelling and lameness of the affected limb. The ultrasound examination evaluated lesion echogenicity and lesion percentage. After the treatment, pain, swelling and lameness were significantly improved by HILT compared with the control group (*p* = 0.023, 0.008 and 0.044, respectively). No significant changes were found in lesion echogenicity degree between both groups in measurements taken during treatment (*p* = 0.188) and after treatment (*p* = 0.070). For lesion percentage reduction, the statistical modelling showed a significant improvement in the HILT group compared with the control group during (*p* = 0.038) and after treatment (*p* = 0.019). In conclusion, HILT promoted analgesic and anti-oedema effects, with visual lameness reduction in horses with tendon and ligament injuries, and reduced lesion percentage but did not influence change in lesion echogenicity.

## 1. Introduction

Tendons and ligaments are important structures in the musculoskeletal system of distal limbs, transmitting large forces between muscles and bones [1]. Tendon injuries occur often in athletic horses, requiring a long rehabilitation process. Injury recurrence is common and negatively affects horse performance [2,3,4]. Tendon injury diagnosis is based on both orthopaedic and ultrasound examination. Ultrasound scanning is a reliable and objective method of assessing the healing process of an injured tendon in response to the treatment methods employed [5,6]. Conservative treatment of tendinopathies includes anti-inflammatory therapy, local cooling and controlled exercise programmes. Several medical and surgical treatments have been proposed as support for tendon repair [7]. However, new therapies aimed at improving the time and quality of tendon repair are still being introduced to veterinary medicine.

Laser therapy has recently been introduced in equine veterinary medicine rehabilitation [8,9,10]. The term LASER is an acronym for Light Amplification by Stimulated Emission of Radiation. It is a device that produces coherent, collimated and monochromatic light through a process of optical amplification based on the stimulated emission of electromagnetic radiation. Laser therapy uses light within the red and infrared parts of the electromagnetic spectrum. Treatment with class 3B lasers (with a power not exceeding 500 mW) is called low level laser therapy (LLLT), whereas the use of therapeutic lasers with a much higher power (class 4 lasers) is called high intensity laser therapy (HILT) [11]. Various studies have shown that laser light has anti-inflammatory, analgesic and antioedematous effects [12,13,14,15]. It can also stimulate fibroblast activity and proliferation [12,16,17] and increase adenosine triphosphate production [18]. Furthermore, during HILT the temperature in the treatment area rises, which increases the intensity of metabolic processes in cells [19].

The application of HILT in equine veterinary medicine has not been widely studied. Recent studies have indicated the effectiveness of HILT for management of medial carpal collateral ligament injury [20] and osteoarthritis [21]. However, those studies involved small datasets and lacked a control group. Therefore, further work is required aimed at identification of HILT effectiveness.

The aim of this study was to evaluate the effects of HILT on injured tendon and ligament repair in horses. The study was conducted to assess the influence of HILT on pain, soft tissue swelling and lameness associated with tendon and ligament injury. The ultrasound analyses focussed on changes in lesion echogenicity and its percentage. It was hypothesized that HILT in the treatment of tendon and ligament injury would reduce pain, swelling and lameness and promote healing.

## 2. Materials and Methods

### 2.1. Animals

The study was based on 26 warmblood performance horses of both sexes, with a mean age of 11.5 years (range 5–24 years), who were clinical patients of the Department of Surgery, Faculty of Veterinary Medicine, Wroclaw University of Environmental and Life Science. Horses were housed separately in stable boxes with the same straw bedding material. They were fed on good quality hay twice a day and had free access to water.

Horses were selected for the study according to the following criteria:Physical evidence of tendon or ligament injury (SDFT, superficial digital flexor tendon; DDFT, deep digital flexor tendon; SL, suspensory ligament), where at least two of the following symptoms were present: pain on palpation, limb swelling, lameness;Presence of tendon or ligament injury with damage of collagen fibres visible in ultrasound scanning;No previous treatment of the present injury.

The study was approved by the Local Ethical Committee for Experiments on Animals (no 39/2018 from 18 April 2018).

### 2.2. Study Design

Horses were divided into a treatment group (= group HILT) and a control group (= group C) by using simple randomization. All horses followed the same 40-day rehabilitation period, which included a controlled activity programme in the form of two 20-min walks a day on hard ground, followed by local cold water therapy consisting of pouring a strong stream of cold water over the tendons and ligaments of the affected limb for 20 min after each walk. Each horse in the HILT group received a series of fifteen HILT treatments. Laser therapy was performed after the first walk and limb cooling, when the skin of the injured limb was dry. Before treatment (BT, day 0), during treatment (DT, day 13–15) and after treatment (AT, day 38–40), a clinical assessment and ultrasound examination were carried out to assess the clinical condition of the affected limb (pain, swelling and lameness) and any changes in the injured tendon and ligament visible in ultrasound scanning (lesion echogenicity and lesion percentage). All horses were randomly assigned for clinical assessment and ultrasound examination.

### 2.3. Clinical Assessment

Physical evidence of pain was scored based on horse response to tendon palpation. The pain scale employed in this study was modified from previous methods [22,23]. Response was scored as follows:No reaction to palpation—no pain (0);Avoidance characterized by muscle contraction with slight limb movement—mild pain (1);Resistance to palpation, where the horse is reluctant to lift the limb or pulls the limb away—moderate pain (2);Violent avoidance characterized by nervous reaction—severe pain (3).

Swelling was assessed by measuring the circumference of the injured limb and the opposite healthy limb. The exact location of the circumference measurement was clipped based on its relationship to the accessory carpal bone and tuber calcanei. Clipped regions were maintained throughout the study to ensure consistency of data collection.

For visual lameness examination, horses were observed during left and right side lunging on a soft surface and straight line walk and trot. Lameness was scored using the American Association of Equine Practitioners (AAEP) scale [24]. Each clinical assessment was performed by the same veterinarian (P.Z.).

### 2.4. Imaging

Three hundred and thirty grey-scale digital images were acquired using a 7.5 MHz linear transducer (7L4s, Mindray Ultrasound Probes, Mindray M5, Mindray Medical, Notting Hill, Australia) To achieve better image quality, an ultrasound probe standoff pad was used. Ultrasound scanning was conducted on weight bearing limbs, and horses were restrained in a quiet place without sedation. Injured tendons and ligaments were visualized in both cross-sectional and longitudinal views. Ultrasonographic images were evaluated to determine tendon cross-sectional area (CSA), lesion CSA, lesion percentage (lesion CSA/tendon CSA × 100%) and lesion echogenicity according to the Genovese scale [25]:Isoechoic (0);Mildly hypoechoic (1);Moderate hypoechoic (2);Anechoic (3).

Images were measured and analyzed by the same veterinarian (P.Z.).

### 2.5. High Intensity Laser Therapy

HILT was performed using a class 4 laser Polaris HP S.(ASTAR, Bielsko-Biała, Poland). with two infrared wavelengths delivered simultaneously: 808 nm (AlGaAs laser with output power 8 W) and 980 nm (InGaAs/AlGaAs laser with output power 10 W). The optical power was calibrated by an external service company. Each horse from the treatment group received a total of fifteen pulsed wave HILT treatments with the same laser parameters, using a protocol that allowed tendons and surrounding tissues time to respond to the therapy. Four treatments were performed initially every 24 h, followed by four treatments every 48 h, then four treatments every 72 h and finally three treatments every 96 h. For the 808 nm and 980 nm wavelengths, different laser parameters were used. For 808 nm, the energy density was 16 J/cm^2^ with power 5 W and frequency 700 Hz. For 980 nm, the energy density was 16 J/cm^2^ with power 4 W and frequency 1000 Hz. Lower parameters than the maximum output power of the device were used to reduce thermal effects and avoid skin burns (especially on pigmented skin in treatment areas). Single HILT application time and total energy dose depended on the treatment area and were calculated by the device, using the manufacturer’s algorithm. The treatment area was clipped (blade size 0.5 mm), and the clipped regions were maintained throughout the study. Laser therapy was conducted on weight bearing limbs, and horses were restrained in a quiet place without sedation. Treatment was carried out with a manual scan and contact technique by the same veterinarian experienced in laser therapy (P.Z.). Scanning was performed in both transverse and longitudinal directions of the limb. The laser hand piece was held perpendicular to the skin surface.

### 2.6. Data Analysis

All statistical analyses were performed using STATISICA v.13 (StatSoft, Inc., Tulsa, OK, USA). For the analysis of qualitative variables assessing the degree of pain, lameness and lesion echogenicity for the two groups, the Pearson chi-square test was used. The Shapiro–Wilk test, Levene’s test and Bartlett’s test were performed to ensure the assumptions of normality and equal variances. One-way analysis of variance (ANOVA) followed by the post-hoc Tukey test were used for parametric continuous data to determine differences in the treatment and control group means in relative swelling. For data that were not distributed normally, the independent non-parametric Mann–Whitney U test was applied to compare the mean values of lesion percentage in the two groups. The intragroup results in lesion percentage measures were analyzed by the Friedman test followed by post-hoc Dunn’s test to compare day 0 (BT), day 13–15 (DT) and day 38–40 (AT). For all tests, the significance level was defined as *p* ≤ 0.05.

## 3. Results

### 3.1. Animals

Twenty six horses were included in this study with 29 tendon and ligament injuries. The following tendinopathies and desmopathies were diagnosed: SDFT (n = 12; 41%), DDFT (n = 8; 28%; including three horses with injury of the accessory ligament of DDFT) and SL (n = 9; 31%; including three horses with injury of SL body and six horses with injury of SL branches). In nineteen horses (73%) injuries involved front limbs, and in seven horses (27%) injuries were located in the hind limbs. Two of the horses were diagnosed with an injury of SDFT and SL in the same limb, and one horse had an injury of SL in both front limbs. Twenty three tendons and ligaments were randomly assigned to group HILT, and six tendons and ligaments were assigned to group C. For the horse with an injury of SL in both front limbs, the left limb was assigned to group HILT and the right limb was assigned to group C. All the horses completed all treatments and measurements. No statistically significant differences were found between group HILT and group C at day 0 in the clinical assessment and ultrasonographic examinations. No abnormalities in behaviour, general condition or appetite were noticed during and after HILT application.

### 3.2. Clinical Assessment

#### 3.2.1. Pain

There was no statistically significant difference between group HILT and group C during treatment in pain response for tendon and ligament palpation scores (*p* = 0.100). After treatment, in group C pain scores were significantly higher compared with group HILT (*p* = 0.023). Furthermore, pain reduction after treatment was statistically significant in group HILT (*p* = 0.001), while in group C no statistically significant pain reduction was observed (*p* = 0.895; Figure 1, Table 1).

#### 3.2.2. Swelling

In the control group, the relative swelling systematically decreased, but the differences were not significant (*p* = 0.667). In group HILT, the relative swelling decreased when comparing the results before and during treatment (6.4 % vs. 4.2%), and this difference was on the border of significance (*p* = 0.072). At the end of the study, when comparing the results before and after treatment, the difference was significant (6.4% vs. 2.4%; *p* = 0.001). Additionally, there was a statistically significant difference between group HILT and group C in relative swelling reduction measured during treatment (4.2% vs. 7.8%; *p* = 0.024) and after treatment (2.4% vs. 6.8%; *p* = 0.008; Figure 2, Table 2).

#### 3.2.3. Lameness

There was no statistically significant difference between the two groups in visual lameness degree during treatment (*p* = 0.596). After treatment, AAEP scores were significantly lower in group HILT compared with group C (*p* = 0.044). In addition, at the end of the study there was a statistically significant difference in lameness reduction in group HILT (*p* < 0.001) but no statistically significant difference in lameness reduction in group C (*p* = 0.970; Figure 3, Table 1).

### 3.3. Ultrasound Examination

#### 3.3.1. Lesion Echogenicity

No significant changes were found between group HILT and group C in lesion echogenicity in measurements performed during treatment (*p* = 0.188) and after treatment (*p* = 0.070; Table 1). However, compared with the baseline there was a significant reduction in lesion echogenicity measured at the end of the study in group HILT (*p* < 0.001) but no significant reduction in the control group (*p* = 0.558; Figure 4, Table 1).

#### 3.3.2. Lesion Percentage

The results of the statistical analysis showed that there was a difference in lesion percentage reduction between the two groups during and after treatment (*p* = 0.038 and 0.019, respectively; Table 3). Additionally, in group HILT the analysis revealed significant differences in lesion percentage reduction when comparing the results before and during treatment (24.1% vs. 20.7%; *p* < 0.001), during and after treatment (20.7% vs. 15.7%; *p* = 0.001) and before and after treatment (24.1% vs. 15.7%; *p* < 0.001). In the control group, a statistical difference was observed only when comparing the results before and after treatment (39.7% vs. 35.7%; *p* = 0.043; Figure 5, Table 3).

## 4. Discussion

The phenomenon of photobiostimulation has been studied since it was discovered [26]. Laser treatment is non-invasive and non-painful, and can be easily administered in primary care settings for a wide range of conditions [27]. The use of HILT in equine veterinary medicine is starting to gain popularity. So far, many experiences of HILT have been reported, but mainly based on clinical cases described by practitioners [9,10,20]. To our knowledge, this is the first study to document the clinical impact of HILT on tendon injury repair in horses where only HILT is used as a main therapy and the study design includes both treatment and control groups.

Previous studies presented by Fortuna et al. [9] and Pluim et al. [10] treated clinical tendinopathies and desmopathies of SDFT, DDFT and SL with HILT, and their conclusion was that HILT considerably reduced the time necessary to obtain clinical and ultrasound resolution of injury. However, in Pluim et al.’s study, HILT was either applied alone or with other therapies. An additional lack of a control group precludes the drawing of any conclusions about direct HILT effects. In Fortuna et al.’s research, horses in the clinical survey were divided into active laser and sham laser groups, but all the horses underwent local immunostimulation injection with *Parapoxvirus ovis*.

Our study confirmed the hypothesis that the application of HILT in a treatment group leads to a significant improvement in the symptoms of pain, limb swelling and lameness that are associated with tendon and ligament injury. Compared with the control group, significant pain and lameness reductions were observed only after cessation of laser therapy, while significant swelling reduction was found both during and after HILT treatment. It can be assumed that the achieved analgesic effect of HILT influenced visual lameness reduction.

Laser electromagnetic radiation transforms prostaglandins into prostacyclin, blocking a mediator of inflammation, which results in an anti-inflammatory effect [28]. The pain relief effect is achieved through nociceptor suppression by change of axon perfusion [29,30]. Finally, laser light induces hyperaemia and activates microvascularization, which result in an anti-oedema effect [31].

Previous studies described the anti-inflammatory, anti-oedematous and analgesic effects of HILT on affected soft tissue and joints [14,15,32]. In our study, the findings of pain, lameness and swelling reduction corroborate those of Pluim et al. [10] and Fortuna et al. [9], but our effects were achieved later. It could be argued that we used much higher pulse mode frequencies (700 Hz and 1000 Hz compared with 29 Hz, 33 Hz and 40 Hz). It is known that low pulsing frequency of laser light results in a better pain control effect [29,33]. In our study, we believed that choosing pulse mode frequencies that worked with high peak power would better induce a healing reaction in damaged tendons. There is no universally accepted HILT protocol for tendinopathies and desmopathies concerning the number of sessions, duration, frequency and dose of energy.

In the present study, the ultrasound evaluation showed a lack of significant changes between the HILT and control groups in lesion echogenicity. Notwithstanding, the interpretation of evolution of the lesion percentage shows significant differences between the groups. Tendon healing can occur intrinsically, by proliferation of endotendon tenocytes, or extrinsically, by migration of cells from the surrounding synovium and sheath [34,35]. Initially, collagen is produced by epitendon cells [36]. Extrinsic compartment involvement in healing is often demonstrated as extensive vascular outgrowth from the peritenon [37], and tenoblasts initiate the repair process through proliferation and migration [38]. HILT increases angiogenesis and fibroblast activity, leading to increased collagen production and improved tensile strength [39]. Although our study did not include histopathological analysis, the ultrasound examination suggests a possible positive influence of HILT on tendon and ligament healing, especially in the early stages of the repair process.

Previous studies in human medicine also describe positive effects of HILT on tendon healing [40,41]. In equine medicine, both Pluim et al. [10] and Fortuna et al. [9] reported significant improvement in tendon injury healing during and after HILT, which was observed much faster than described in the literature for other treatments. Additionally, both studies described HILT as a safe procedure. None of the horses included in our study experienced skin burns or increases in swelling, pain or lameness after HILT application. Therefore, HILT was classified as a safe procedure, which is well tolerated by horses and does not require pharmacological sedation.

Limitations of the present study include a small control group and wide age range in both groups. It would be ideal to have a larger number of horses with tendinopathies and desmopathies to observe the healing process without HILT, subjected only to the rehabilitation program. However, it was important to perform clinical research on horses with possible similar clinical and ultrasound appearance, free of other treatments and additional medication. This study assessed the short-term effect of HILT in tendon and ligament injury treatment. There is a need for long-term follow-up, such as assessing time to return to controlled exercise, time to return to full performance level and re-injury rates.

## 5. Conclusions

In conclusion, HILT treatment resulted in significantly reduced pain, limb swelling and lameness associated with tendon and ligament injures and may be a non-invasive and nonpharmacological way to initially treat clinical signs of tendon and ligament injury. In addition, in ultrasound assessment HILT significantly reduced injury percentage, but lesion echogenicity did not improve in comparison with the control group. Therefore, HILT can be useful as a supportive therapy for healing of tendon and ligament injuries in horses.

## Figures and Tables

**Figure 1 animals-10-01327-f001:**
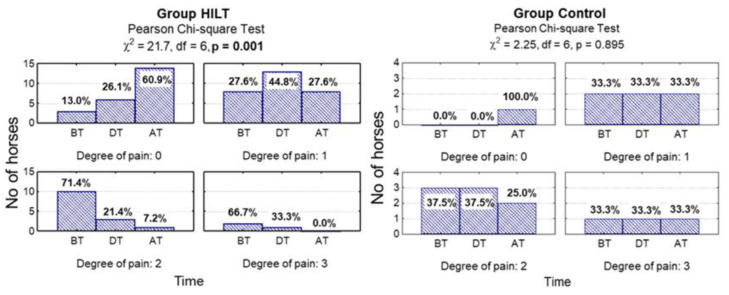
Number (percentage) of horses in groups differing in examination time and degree of pain in the high intensity laser therapy (HILT) group and control (C) group, and the results of the independence test. BT, before treatment (day 0); DT, during treatment (day 13–15); AT, after treatment (day 38–40).

**Figure 2 animals-10-01327-f002:**
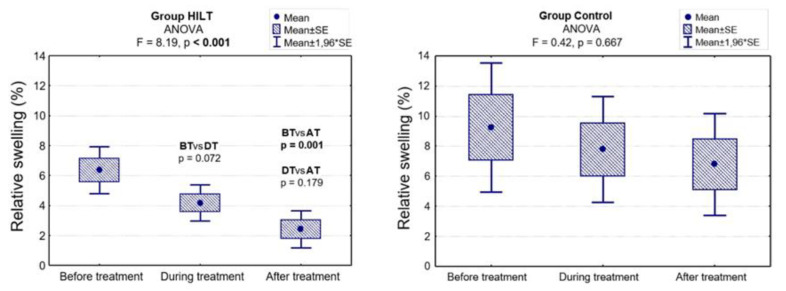
Relative swelling before, during and after treatment in group HILT and group C, and the ANOVA results. BT, before treatment (day 0); DT, during treatment (day 13–15); AT, after treatment (day 38–40).

**Figure 3 animals-10-01327-f003:**
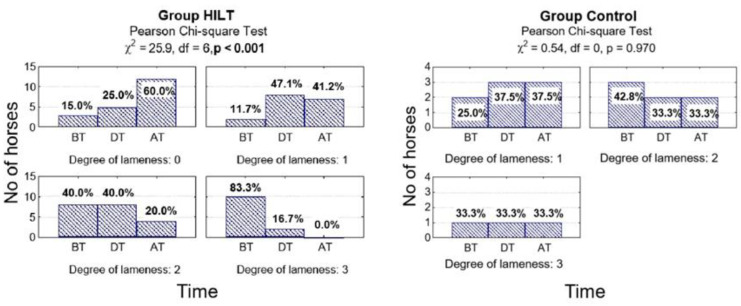
Number (percentage) of horses in groups differing in examination time and degree of lameness on the American Association of Equine Practitioners (AAEP) scale in the HILT group and control group, and the results of the independence test. BT, before treatment (day 0); DT, during treatment (day 13–15); AT, after treatment (day 38–40).

**Figure 4 animals-10-01327-f004:**
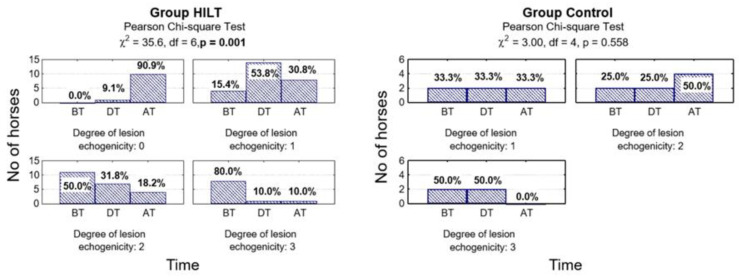
Number (percentage) of horses in groups differing in examination time and lesion echogenicity degree in the HILT group and control group, and the results of the independence test. BT, before treatment (day 0); DT, during treatment (day 13–15); AT, after treatment (day 38–40).

**Figure 5 animals-10-01327-f005:**
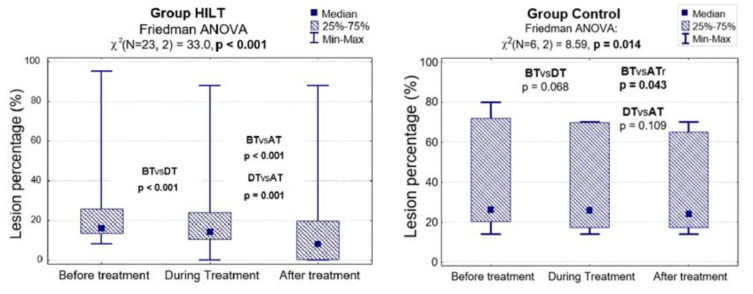
Lesion percentage before, during and after treatment in group HILT and group C, and the Friedman ANOVA results. BT, before treatment (day 0); DT, during treatment (day 13–15); AT, after treatment (day 38–40).

**Table 1 animals-10-01327-t001:** The effect of HILT on pain, lameness and lesion echogenicity for horses from high intensity laser therapy (HILT) group and control (C) group.

Clinical Assessment and Ultrasound Examination	Degree of Pain			Degree of Lameness			Degree of Lesion Echogenicity		
	Group HILT N = 23	Group C N = 6	HILT vs. C *p*-Value	Group HILT N = 23	Group C N = 6	HILT vs. C *p*-Value	Group HILT N = 23	Group C N = 6	HILT vs. C *p*-Value
Degree	n (%)	n (%)	n (%)	n (%)	n (%)	n (%)	n (%)	n (%)	n (%)
Before treatment			0.772			0.254			0.667
0	3 (13.0)	0 (0.0)		3 (13.0)	0 (0.0)		0 (0.0)	0 (0.0)	
1	8 (34.8)	2 (33.3)		2 (8.7)	2 (33.3)		4 (17.4)	2 (33.3)	
2	10 (43.5)	3 (50.0)		8 (34.8)	3 (50.0)		11 (47.8)	2 (33.3)	
3	2 (8.7)	1 (16.7)		10 (43.5)	1 (16.7)		8 (34.8)	2 (33.3)	
During treatment			0.100			0.596			0.188
0	6 (26.1)	0 (0.0)		5 (21.7)	0 (0.0)		1 (4.3)	0 (0.0)	
1	13 (56.6)	2 (33.3)		8 (34.8)	3 (50.0)		14 (60.9)	2 (33.3)	
2	3 (13.0)	3 (50.0)		8 (34.8)	2 (33.3)		7 (30.5)	2 (33.3)	
3	1 (4.3)	1 (16.7)		2 (8.7)	1 (16.7)		1 (4.3)	2 (33.3)	
After treatment			0.023			0.044			0.070
0	14 (60.9)	1 (16.7)		12 (52.2)	0 (0.0)		10 (43.5)	0 (0.0)	
1	8 (34.8)	2 (33.3)		7 (30.4)	3 (50.0)		8 (34.8)	0 (50.0)	
2	1 (4.3)	2 (33.3)		4 (17.4)	2 (33.3)		4 (17.4)	2 (33.3)	
3	0 (0.0)	1 (16.7)		0 (0.0)	1 (16.7)		1 (4.3)	4 (66.7)	
Degree of reduction *p*-value	**<0.001**	0.895	-	**<0.001**	0.970	-	**<0.001**	0.558	-

N, total number of horses in groups; n, number of horses in each group.

**Table 2 animals-10-01327-t002:** The effect of HILT on relative swelling (median and IQR) for horses from group HILT and group C.

Relative Swelling	Group HILT N = 23	Group C N = 6	HILT vs. C *p*-Value
Before treatment (%):			0.144
Me [Q1; Q3]	6 [4; 11]	8 [6; 15]	
Min–Max	0–15	2–16	
During treatment (%)			0.024
Me [Q1; Q3]	3 [2; 5]	7 [5; 11]	
Min–Max	0–12	2–14	
After treatment (%)			**0.008**
Me [Q1; Q3]	2 [0; 3]	6 [4; 8]	
Min–Max	0–12	2–14	
ANOVA: *p*-value	<0.001	0.667	-

**Table 3 animals-10-01327-t003:** The effect of HILT on lesion percentage reduction (median and IQR) for horses from group HILT and group C.

Lesion Percentage (%)	Group HILT N = 23	Group C N = 6	HILT vs. C *p*-Value
Before treatment (%)			0.085
Me [Q1; Q3]	16 [13; 26]	26 [20; 72]	
Min–Max	8–95	14–80	
During treatment (%)			0.038
Me [Q1; Q3]	14 [10; 24]	26 [17; 70]	
Min–Max	0–88	14–70	
After treatment (%)			**0.019**
Me [Q1; Q3]	8 [0; 20]	24 [17; 65]	
Min–Max	0–88	14–70	
ANOVA: *p*-value	<0.001	0.014	-

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
