# Peer review of "Effects of High Intensity Laser Therapy in the Treatment of Tendon and Ligament Injuries in Performance Horses"

_animals, 2020, doi:10.3390/ani10081327_

Round 1
Reviewer 1 Report
Thank you for your interesting and well written paper. I know I have made quite a few comments, but these are all only very minor amendments, additions or questions.
Line 20 – missing space between sentences
Line 35 – Would it be worth adding into the abstract who performed your clinical assessments? Was this a vet and were all assessments completed by the same person?
Line 37 – Could you give some indication of directionality to these results? i.e. were these parameters reduced in your treatment group? This is not made clear as you have just said “differences” and maybe this is worth making clear in the abstract earlier rather than just at the end of the abstract.
Line 50 – and also career longevity? No need for a comma between both and orthopaedic
Line 54 – when you say tendon support, could you make it clearer that you are supporting the repair of the tendon?
Line 66 – state ATP in full
Line 68 – in horses?
Line 81 and 94 – Is there a reason why you used simple randomisation? With such a big age range would it not have been useful to pair horses by age and assign one of each age group to either control or treatment groups?
Line 82 – were the horses all housed on the same stable bedding type?
Line 97 – How were the HILT sessions timed throughout the rehabilitation programme and were these standardised across the sample?
Line 98 – Your assessments seem to have been spread over a few days i.e. day 13 – 15. Were the different groups randomly assigned which day the individuals were assessed in or were all in one group assessed on one day e.g. all control horses assessed on day 13 and all treatment group on day 15? I think it is important to make this clear
Line 117 – do you mean that all assessments were completed by the same person and was this person a vet?
Line 125 – were all of these images analysed by the same person?
Line 138 – not quite clear what you mean by “own protocol”. Do you mean each horse’s own protocol?
Line 149 – I take it that the person using HILT was experienced in this treatment? I think this is worth stating
Line 169 – I assume that it was a welfare concern for the horse that made you have a considerably smaller control group than treatment group?
Line 196 – do you think that the differences in the spread of data between groups may have had an influence?
Line 237 – same question as above
Line 250 – not sure “a lot of” is the best term here
Line 298 – do you think the age range of your horses may have been a limitation as well?
Reviewer 2 Report
Effects of High Intensity Laser Therapy in the treatment of tendon injuries in performance horses.
General considerations
The title of the paper is “Effects of High Intensity Laser Therapy in the treatment of tendon injuries in performance horses.” but in the reported cases Authors included treatment on SDFT, DDFT but also some SL. Consequently describing the work, every sentence that contains the word tendon must be accompained with ligament and when tenopathy is considered, also desmopathy must be.
Again, equine flexor tendons and SL differ each other in structure, healing process and general response to many treatment methods.
Personally and confidentially I consider that the inclusion of SLs, as reported in the text, could complicate the setting of the study design. To be clearer, I’m not against it but I suggest to consider the opportunity to underline in the paper possible problems in uniformity of the pathologic data including tendons and ligaments.
Special considerations:
2. title Effects of High Intensity Laser Therapy in the treatment of tendon injuries in performance horses – not only tendon but also ligaments (SL)
19. “26 horses” – modify in “Twentysix horses”
31. “26 horses” – modify in “Twentysix horses”
167. “26 horses” – modify in “Twentysix horses”
19-20. “to treated with HILT and a non-treated group” – correct for example with “randomly assigned to a HILT treated or to a non-treated group”.
20. “non-treated group.Horses from..” – better “Each horse from..”
21. “The clinical and ultrasound” – delete “The”
25. “not influence” – correct “not influenced”
31. “assigned to treated with HILT and a non-treated group.” Same of line 19 “randomly assigned to a HILT treated or to a non-treated group”.
49. “Injury recurrence is common, which negatively affects” – change “and negatively affects”
51- 53. “Conservative treatment of tendinopathies includes anti-inflammatory therapy, local cooling and controlled exercise programmes. Ultrasound scanning is a reliable and objective method of assessing the healing process of an injured tendon in response to the treatment methods employed [5,6]. Several medical and surgical treatments have been proposed for tendon support [7].” – inconclusive sentences, probably better if the Authors complete the brief review on diagnosis and then focus on the treatment, inverting the two sentences : “ultrasound scanning” .. and then .. “conservative treatment”.
57. “Laser therapy has been introduced in equine veterinary medicine rehabilitation recently [8]”. Probably better to insert more bibliographic citations considering the laser therapy on tendons, not only one. (see eventually citations in Pluim et al.)
72. “effects of HILT on tendon injury treatment in horses” – better substitute with “injured tendon healing” or “repair”.
81 being a clinical patients – “a” is not necessary in this sentence.
93. “divided into a treatment group (=group HILT) and control group” – better “a control group”
97. “Group HILT had a series of fifteen HILT treatments” – better change in “Horses (or Each horse) in the HILT group received a series of fifteen HILT treatment.”
117-118. “Each clinical assessment was performed by the same person.” – better to declare the acronim of the author who did the assessment between brackets.
170. “the left limb was assigned to group HILT and right” – better “and the right limb”
189. “In the group HILT the relative swelling already decreased” – what does it means “already” in this context? (same and less comprehensible in line 192)
216-219. The content of this paragraph could be better described in the Results at the paragraph 3.1, to present to a reader the complete the records, giving the possibility to understand and interpret further data in the paper.
216. not only “tendinopaties” but also “desmopathies”. See the first consideration.
252. “impact of HILT on tendon injury treatment in horses” - better change treatment with “healing” or “repair” because in the acronim HILT, therapy is included and its effect or impact could be on repair or healing, not on the treatment itself.
252. it is limiting to consider HILT as an additional therapy. Probably it could be considered the main one toghether with rehabilitation. A more suitable sentence could give importance to the treatment method adopted.
